# Strain Hardening in an AZ31 Alloy Submitted to Rotary Swaging

**DOI:** 10.3390/ma14010157

**Published:** 2020-12-31

**Authors:** Zuzanka Trojanová, Zdeněk Drozd, Kristýna Halmešová, Ján Džugan, Tomáš Škraban, Peter Minárik, Gergely Németh, Pavel Lukáč

**Affiliations:** 1Faculty of Mathematics and Physics, Charles University, Ke Karlovu 5, 12116 Praha 2, Czech Republic; ztrojan@met.mff.cuni.cz (Z.T.); Zdenek.Drozd@mff.cuni.cz (Z.D.); Tskraban@seznam.cz (T.Š.); Peter.Minarik@mff.cuni.cz (P.M.); gergely1227@gmail.com (G.N.); lukac@met.mff.cuni.cz (P.L.); 2COMTES FHT a.s., Průmyslová 995, 33441 Dobřany, Czech Republic; jan.dzugan@comtesfht.cz

**Keywords:** magnesium alloy AZ31, grain size, strain hardening, twinning, dislocation density, residual stresses

## Abstract

An extruded magnesium AZ31 magnesium alloy was processed by rotary swaging (RSW) and then deformed by tension and compression at room temperature. The work-hardening behaviour of 1–5 times swaged samples was analysed using Kocks-Mecking plots. Accumulation of dislocations on dislocation obstacles and twin boundaries is the deciding factor for the strain hardening. Profuse twinning in compression seems to be the reason for the higher hardening observed during compression. The main softening mechanism is apparently the cross-slip between the pyramidal planes of the second and first order. A massive twinning observed at the deformation beginning influences the Hall-Petch parameters.

## 1. Introduction

In theoretical models, the development of plastic deformation is usually described on the base of the development of dislocation substructure. This approach assumes that the plastic deformation is determined by one structural parameter *S*, which describes the current structural state. The stress, *τ*, necessary for ongoing plastic deformation in a slip plane is calculated at temperature *T*, by the following kinetic equation:
(1)$$\tau =(S,\dot{\gamma },T)$$which corresponds to the dynamic equilibrium of the structure at a strain rate $\dot{\gamma }$. In the literature, theoretical models using several structural parameters *S_1_*, *S_2_*, *…*, *S_n_* have been used. Achieving a saturated state is then an analogy of the steady creep, i.e., the state when the hardening and softening processes are in equilibrium. In one parametric models, only one structural parameter is decided for the development of plastic deformation [1,2,3,4,5,6]. Two and multiparameter models, taking into account other aspects of the microstructure, consider that each structural parameter has its stationary value which may be achieved after a certain deformation. The deformation process is then controlled by the slowest of them [7,8,9,10]. The suitable parameter defining the structure *S* is the dislocation density, *ρ*. The structure development during plastic deformation is determined by the evolution equation:(2)$$\frac{dS}{d\gamma }=\frac{d\rho }{d\gamma }=f\left(\dot{\gamma },\;T,\;S\right)$$

The evolution equation, which describes the development of the dislocation substructure with time (or strain, *γ*, both quantities are proportional) contains generally two types of terms: the production (hardening) term and the annihilation (recovery) term:(3)$$\frac{d\rho }{d\gamma }=\;{\left(\frac{d\rho }{d\gamma }\right)}_{h}-\;{\left(\frac{d\rho }{d\gamma }\right)}_{r}$$

The flow stress, *τ*, necessary for plastic deformation, is obtained by the known relationship:*τ* = *τ*_0_ + α*Gb**ρ*^1/2^(4)

Here, *τ**_0_* is yield strength, *G* is shear modulus, *b* is the Burgers vector of dislocations and α a numerical parameter, considering the character of mutual dislocation interactions.

The deformation behaviour is usually described by the work-hardening rate *θ* = *d**τ**/d**γ*. Back to the stress, *τ*, it is possible simply return the expression:(5)$$\tau =\overset{\gamma }{\underset{0}{\int }}\theta \left(\gamma \right)d\gamma$$

The work-hardening coefficient, *Θ*, of polycrystals *Θ* = d*σ*/d*ε* is related to the work-hardening coefficient, *θ*, for single crystals by the relationship:*Θ* = *M*^2^*s**θ*(6)
where *M* is the Taylor factor and s a factor describing the influence of the thermal activation, while σ = *M**τ* and *ε*
*=*
*γ*/*M.*

The work hardening processes in fcc metals have been intensively studied and are relatively well explored [1,2,3]. The role of various hardening and softening processes at different temperatures is analysed in the most general model by Lukáč and Balík [2], who formulated an evolution equation for the dislocation density with two production terms and two annihilation terms: multiplication of dislocations by both impenetrable obstacles and forest dislocations and the process of dislocation annihilation due to cross-slip and climb of dislocations. Contrary to fcc metals, the work hardening rate has been studied less frequently in hcp metals because of their strong plastic anisotropy and shortage of easy glide slip systems [4,5,6,7]. Máthis et al. [8] applied the Lukáč and Balík model to cast Mg-Al alloys and AM20 magnesium alloy samples. They estimated that the main strengthening contribution to the storage of dislocations is the forest dislocations lying in the nonparallel slip planes. On the other hand, the cross-slip plays at room temperature only a negligible role, contrary to the dislocation climb, which may be an effective softening process. At elevated temperatures, dynamic recrystallization can be observed. Similar results were found applying this model to other cast magnesium alloys [6]. Strain hardening in pure magnesium due to twinning was studied by Cáceres et al. [9]. They estimated only a small influence of twinning on dislocation motion in randomly oriented polycrystals. Generally, the mentioned theories consider dislocation density, dislocation evolution and annihilation. If textured hcp materials are loaded under orientation of the stress axis causing profuse twinning or detwinning, the stress-strain curves exhibit a sigmoidal character. Oppendal et al. [10] pointed out the limitation of current hardening models in predicting plastic anisotropy due to twinning in hcp metals. They used viscoplastic self-consistent modelling to explain the experimentally observed latent hardening parameters. Coupling between slip, twinning, texture, and stress state was considered. An ECAPed ZK60 alloy was studied in [11] with the aim to reveal the influence of grain size, dislocation density, and twinning on the strength. They estimated an increase in the dislocation density up to the second pass through the ECAP tool. Observed storage of dislocations is due to increased twin boundaries and subgrain formation during high temperature deformation in the ECAP tool. The role of twinning and grain size in the strain hardening in tension and compression of pure magnesium, randomly oriented, was studied by Cáceres and Blake [11]. Like in fcc metals they tagged the forest dislocation accumulation as the main contribution to the strain hardening. The influence of stacking faults on the work hardening processes in as-cast ZK10Yb alloy was studied in [12]. Stacking faults and deformation twins were found as the main obstacles for dislocation motion. Zhang et al. [12] studied the work hardening of as-cast Mg-1Zn-0.4Zr-0.2Yb alloy deformed in tension at room temperature. A small addition of Yb caused a decrease of the stacking fault energy of basal dislocations. The authors described the observed high work hardening due to increased activity of basal and non-basal dislocations and twinning. According to the authors of this paper, the high work hardening values may be described by a higher splitting of basal dislocations and higher stress necessary for the withdrawal of screw dislocations and their passage into the cross-slip plane.

Mechanical properties of rolled sheets of an AZ31 alloy were investigated in [13] with the aim to reveal the possible influence of the grain size, texture, and twinning. The work hardening rapidly increased with the decreasing grain size. On the other hand, the authors of [13] estimated only a negligible influence of the (0002) pole intensity on the work hardening rate. Similarly, del Valle et al. studied the grain size effect on work hardening and ductility in AZ31 and AM60 magnesium alloys prepared by ECAP and hot rolling [14]. The importance of non-basal slip for the recovery process(-es) at room temperature in an ECAPed AZ31B was analysed by Koike et al. [15]. The effect of Mn on the tensile properties of the extruded Mg-1Sn based alloys was investigated at room temperature [16]. The authors estimated that the yield stress asymmetry decreases with increasing Mn content and so the work hardening ability. They found that the Mn precipitates refined substantially the microstructure. The observed decrease of the work hardening coefficient and the lower twinning intensity were explained by the smaller grain size.

Knezevic and co-workers studied the influence of deformation twinning in AZ31 on various aspects of plastic deformation, tension-compression asymmetry, work hardening rate, and the evolution of crystallographic texture [17]. Tension-compression asymmetry of rolled Mg-Y alloys was analysed in [18] at 4 K, 78 K and room temperature. The authors highlighted the influence of solute atoms on the activity of various slip systems and mechanical twinning. The influence of texture on the work hardening of Mg-2Zn-0.1Ca sheet has been reported by Shou et al. [19]. The hardening behaviour strongly depends on the orientation of the stress axis with respect to the rolling direction and is governed by the distribution of Schmid factors for basal slip. The low values of the Schmid factor led to the high work hardening rate and higher activity of twinning in the transversal direction. The effect of Zn content on the work hardening behavior of Mg-1Gd sheets was studied in [20]. The Zn addition increased the hardening capacity. This effect depends on solid solution hardening, texture evolution, and precipitation hardening. The texture development and hardening evolution along three principal directions was studied in samples of AZ31 alloy submitted to ECAP deformed in compression [21]. The tendency to twinning in compression tests was found to be higher in samples stressed in the extrusion direction and decreasing in samples with the stress axis perpendicular to the extrusion direction. Trojanová et al. studied the hardening and softening processes in an AX41 (Mg-Al-Ca) alloy reinforced by short Saffil fibres [22]. They estimated that the short ceramic fibres and twin boundaries are the main obstacles for dislocation motion. Recovery processes observed at elevated temperatures are much stronger compared to the unreinforced alloy. This is due to dislocation pile-ups, formed by twin boundaries and fibres, which may stimulate the climb of dislocations. Recently, Vinogradov and co-workers [23] have presented a phenomenological model of strain hardening for magnesium materials with various grain sizes and textures. They considered the interplay between dislocation slip and mechanical twinning in the deformation of materials with twinning-mediated plasticity. Two internal variables are considered: the total dislocation density and the twin volume fraction. The model was constructed under assumptions regarding the stress-driven twinning kinetics and dislocation-twin interactions. Constitutive equations governing the strain hardening behaviour were obtained and solved numerically.

Results presented in the literature showed that the development of plastic deformation in hexagonal metals, particularly in magnesium materials, is a complex problem where the solute atoms, temperature, grain size, texture, and anisotropy of these materials play the deciding role. Solute atoms can form precipitates as effective obstacles for dislocation motion and, on the other hand, may influence the critical resolved shear stress in various slip systems in different ways. The activities of all these parameters were studied exclusively in the cast alloys and alloys after ECAP or rolling processing. Although the texture developed in magnesium alloys during rotary swaging is often similar to the texture of extruded materials, the deformation behaviour may be different [24].

Rotary swaging is an incremental metal forming process for the reduction of cross-sections of bars, tubes and other cylindrical workpieces. It belongs to the category of net-shape-forming processes, where the formed workpiece is obtained without, or with only a minimum amount of processing and cutting. The forming dies of the swaging machine are arranged concentrically around the workpiece. Set of dies (from two to eight) performs small, high-frequency, simultaneous radial movements (oscillations) and applies compressive force onto the enclosed workpiece. With every stroke of the die, the material begins to flow and is formed with great precision. Parts produced by rotary swaging are used in different applications, such as the automotive industry where it is used for components, like axes, steering spindles and gear shafts.

In the present paper, the work hardening behaviour of magnesium alloy AZ31 was studied in tension and compression tests. The experiments were aimed to find and discuss the influence of the rotary swaging process on hardening and softening mechanisms in various stages of straining.

## 2. Materials and Methods

The AZ31B (3 wt%Al-1 wt%Zn-0.2 wt%Mn-balance Mg) bars were supplied by Luoyang Magnesium Gurnee Metal Material (Luoyang, China). Extruded rods were subjected to a repeated rotary swaging process. Rotary swaging machine with four-split die from HPM (Heinrich Müller Maschinenfabrik GmbH, Pforzheim, Germany) was used for the experiments. The swaging process was performed five times; an original diameter of 20 mm was reduced to 8 mm. The rods were preheated at a temperature of 450 °C. Samples prepared from the particular rods were depicted as 1 × swaged (SW1), 2 × swaged (SW2), and analogously SW3, SW4, SW5, and the original rod SW0. The microstructure was studied using an Auriga Compact scanning electron microscope (Carl Zeiss, Jena, Germany) equipped with an EBSD camera (EDAX LLC, Draper, UT, USA) and the OIM 7.3 software. Texture analysis was performed using series expansion of generalized spherical harmonics with the axial symmetry condition. Tensile and compression tests were performed at room temperature in an Instron 5882 testing machine (Instron, High Wycombe, UK) at an initial strain rate in the order of 10^−3^ s^−1^. The characteristic stresses as tensile/compression stress (TYS/CYS) and ultimate tensile strength (UTS) and peak compression strength (PCS) were estimated as a true stress at the strain ε= 0.002 and the maximum true tensile/compressive stress. The strain hardening coefficient was calculated as the derivative of the stress—strain curve as Θ=dσ/dε. Cylindrical samples for the thermal expansion measurements have a length of 50 mm and a diameter of 6 mm. Samples were cut from the swaged rods so that the longitudinal axis of the samples was parallel to the extrusion direction. The measurements of sample elongation were performed in an argon atmosphere using a 410 dilatometer (NETZSCH-Gerätebau GmbH, Selb, Germany) over a temperature range from room temperature to 400 °C with heating and cooling rates of 0.9 K/min.

## 3. Results

### 3.1. Microstructure and Texture

The inverse pole figure (IPF) taken from the section perpendicular to the extrusion direction (ED) is shown in Figure 1a and from the section parallel to the ED in Figure 1b. Big non-recrystallised grains are surrounded by the small ones which originated by the partial recrystallisation during the hot extrusion. The grain size distribution is significantly bimodal; the big grains exhibit the grain size ~300 µm, while the small grain size is only 20 µm. The grain size of the samples is reported in Table 1.

EBSD orientation maps estimated for swaged samples are shown in Figure 2. Stepwise swaging refined microstructure. The grain size decreased from 60 µm (SW1) to 16 µm in the five times swaged sample, as shown in Table 1.

The extruded sample exhibited a typically strong $\left\{10\bar{1}0\right\}$ fibre texture combined a with weaker $\left\{11\bar{2}0\right\}$ fibre component (Figure 3). Subsequent rotary swaging resulted in a small decrease in strength of the $\left\{10\bar{1}0\right\}$ fibre component. In the meantime, an increase in the RSW count caused continuous suppression of the $\left\{2\bar{1}\bar{1}0\right\}$ fibre component, as shown in Figure 4.

### 3.2. Work Hardening Coefficient

The work hardening coefficients, *Θ*, depending on strain, are reported together with the stress-strain curves for tension in Figure 5 and for compression in Figure 6. The work hardening coefficients were calculated for stresses higher than TYS/CYS. In Figure 5 and Figure 6 negative values of *Θ* are cut, i.e., only values up to the maximum stress are presented. The shape of stress-strain curves is different for tension and compression. This difference is reflected in the *Θ**-**ε* curves; the *Θ* values estimated for tension rapidly decrease in the small strain range up to 0.02–0.03, then the further decrease is gradual up to zero value. According to Considère criterion, the deformation is stable up to the point where *Θ*
*=*
*σ*. When inequality *Θ*
*<*
*σ* is valid, the strain localisation (necking) occurs.

The strain dependences of *Θ*, estimated in compression, have a completely different character. The work hardening coefficient rapidly decreases in the region of small strains. Approaching a minimum, it steeply increases up to the maximum which is followed by a decrease to zero. Moreover, in this case it is reasonable to take into account only the values *Θ*
*>*
*σ*. Local maxima of *Θ* were observed in all studied samples. These maxima are connected with the rapid stress increase at strains in the interval of 0.04–0.08. This effect is especially strong in the SW0 sample, where the influence of bigger grains strongly influenced the shape of the work hardening curve.

### 3.3. Thermal Expansion

Thermal strains were measured in the temperature region from room temperature up to 400 °C with the aim to estimate possible residual thermal stresses in samples after the thermomechanical treatment. The sample elongation vs. temperature plot measured while heating and cooling is shown in Figure 7 for the SW3 sample as an example of such measurement. It is obvious that the heating and cooling branches follow a different course. Some shortening was observed in all samples. This shortening is a consequence of a relaxation of elastic stresses existing in the samples after rotary swaging. In the next thermal cycle, the heating and cooling branches were found to be identical.

The course of these residual stresses depending on temperature is shown in Figure 8 for three samples. It is obvious that the permanent shortening of samples depends on the RSW counts. The permanent residual strains, estimated after the first heating-cooling cycle, are done in Table 2 for all samples.

## 4. Discussion

### 4.1. Strain Hardening

In the previous paper [24], the acoustic emission (AE) generated during tension and compression was measured. A massive AE was observed, while deformed in tension, at the very beginning of plastic deformation. The broaden maximum of AE became narrower with increasing number of swaging steps. Such a result is different from results usually estimated where the intensive AE is observed only at the vicinity of the yield point [25]. The AE generated during deformation in compression exhibited other characters with three maxima along the stress-strain curves. The first maximum, detected at the beginning of plastic deformation near the yield stress, was followed by the second one. Achieving zero at the strain ~3% the AE activity again rapidly increased up to the third maximum. The width of this third maximum decreases with increasing number of swaging steps. This third maximum lies near the saddle point on the stress-strain curve followed by the rapid stress increase. It is commonly accepted that AE is connected with irreversible changes in the microstructure such as movement of larger dislocation ensembles [26], annihilation of dislocation [27] or crack propagation [28]. Very effective source of acoustic emission is mechanical twinning [29]. It is responsible for the massive AE observed at the beginning of plastic deformation. The different behaviour of AE in compression is due to the growth of twins generated at the beginning of plastic deformation. This growth generates no AE signal and it is the reason for the rapid decrease of AE up to the point when the whole sample volume is twinned. Plastic deformation starts with activation of extension twins $\left\{10\bar{1}2\right\}\;<10\bar{1}1>$. In the textured samples (0001) basal planes are oriented parallel to the extrusion direction while the $\left\{11\bar{2}0\right\}$ prismatic planes rotate around the <c> axis which is perpendicular to the extrusion direction. The twins’ growth in compression is possible due to the geometry of twinning where six variants of twins are formed in tension while in compression only two.

The flow stress of polycrystalline materials depends on the dislocation structure, represented by the mean dislocation density, *ρ*, *σ∝ρ**^1/2^*. During plastic deformation, a part of the dislocation is stored, new obstacles for dislocation motion may be formed, and the slip length is restricted. The mean free path of dislocations, *Λ*, is proportional to the spacing between dislocations *L_d_ =*
*ρ**^−1/2^*:*Λ* = *βL_d_* = *βρ**^−1/2^*(7)
where parameter, *β*, was found in [8] for Zr-Sn alloy to be *β* = 25. The hardening term in the evolution Equation (3) may be written as:(8)$${\left(\frac{d\rho }{d\gamma }\right)}_{h}=\;\frac{\rho^{1/2}}{\beta b}$$

The loss term in Equation (3) is calculated by the mutual annihilation of dislocations in the recovery process(es):(9)$${\left(\frac{d\varrho }{d\gamma }\right)}_{r}=\frac{L_{r}}{b}\rho$$
here *Lr* is the average length of the dislocation segment annihilated in an elementary recovery event. Differentiating relationship (4) and combining with Equation (8), we obtain:(10)$$\left(\tau -\tau_{0}\right)\theta_{h}=\frac{1}{2}{\left(\alpha Gb\right)}^{2}\frac{d\rho }{d\gamma }$$

Equation (10) can be rewritten for polycrystals as:(11)$$\left(\sigma -\sigma_{Y}\right)\frac{d\sigma }{d\varepsilon }=\left(\sigma -\sigma_{Y}\right)\left(\Theta_{h}-\Theta_{r}\right)$$
where for the work hardening coefficient of polycrystals, *Θ*, is done as *Θ** = d**σ**/d**ε* and, *σ**_Y_*, the yield strength.

The evolution of the hardening process was analysed using Kocks-Mecking plots (*σ*
*−*
*σ**_Y_*). *Θ* vs. (*σ*
*−*
*σ**_Y_*) [30]. These dependences, calculated from the stress-strain curves, are plotted for tension and compression together for each swaging state (see Figure 9). It is obvious that the work hardening rate is much higher for the curves of samples deformed in compression excluding a small interval at the beginning of deformation up to *(**σ*
*−*
*σ**_Y_**)* ~50 MPa, where the work hardening coefficient was found lower for the samples deformed in compression. This stress interval corresponds to the twin thickening period where the dislocation strain is insignificant. It is interesting to note that the dependence maximum is for all samples at similar values of *(**σ*
*−*
*σ**_Y_**)* ~200–250 MPa. Inverse pole figures shown in Figure 1, Figure 2 and Figure 3 indicate that only the minimum basal plane has the Schmid factor high enough for the effective slip of <a> dislocations. Máthis et al. found the critical resolved stress for nucleation and formation of extension twins in polycrystalline Mg, randomly oriented 15–40 MPa in tension and 10–30 MPa in compression [31]. Extension twins $\left\{10\bar{1}2\right\}$
$<10\bar{1}1>$ reorient the basal plane by 86.3° and in the case of $\left\{10\bar{1}1\right\}$
$<10\bar{1}2>$ contraction twins with 56.2° with respect to the parent lattice [32,33]. Koike found the critical stress for the formation of contraction twins in Mg alloys to be 76–153 MPa, which is a value much higher compared to the stress necessary for extension twinning [34].

The reoriented basal planes are nearly perpendicular to the stress axis, i.e., the Schmid factor for these planes is low. On the other hand, the pyramidal planes of the first and second order are oriented much favourable. Lou et al. found the critical resolved shear stress in the AZ31 alloy for <a> dislocations in the $\left\{11\bar{2}1\right\}$ pyramidal planes 40 MPa and for <c + a> dislocations in the $\left\{11\bar{2}2\right\}$ pyramidal planes of the second order 45–81 MPa [35]. The maximum of the work hardening coefficient in compression is observed at the stress approximately 300 MPa. It is due to extensive twinning which is manifested by an intensive AE signal [24]. Narrow, secondary and compression twins are generated causing the fast increase of the strain hardening. The basal <a> dislocations are the forest dislocations for pyramidal slip. The interaction between glissile basal <a> and <c + a> pyramidal leads to the formation of sessile dislocations which present extra strong obstacles for dislocation motion [36]:(12)13211¯00001+132¯113211¯2→0001211¯0glissile   glissile   sessile

The resulting sessile <c> dislocation with the Burgers vector parallel to the c axis is not able to glide in any plane. Another reaction of two <c + a> dislocations form a sessile dislocation in the pyramidal plane of the first order:(13)13211¯00001+1311¯231122¯→1312¯13112¯1glissile   glissile   sessile

The work hardening process can be understood as an obstruction of the dislocation motion. The obstacles may be of two types: non-dislocation obstacles and forest dislocations or sessile dislocations which may be formed by the reaction between dislocations. The mean free path of dislocation, Λ, incorporates various terms; it can be written as a combination of several contributions:(14)$$\frac{1}{\Lambda }=\frac{1}{\Lambda_{GB}}+\frac{1}{\Lambda_{twin}}+\frac{1}{\Lambda_{dis}}$$
where ΛGB, Λ*_twin_* are the average grain and twin boundary distances, respectively, Λ*_dis_* is the distance between obstacles of the dislocation type. Accumulation of dislocations on both forest dislocations and the obstacles formed in dislocation reactions is the reason for the observed significant hardening. Knezevic and coworkers assumed that the extension twins in AZ31 sheet are not very effective in strain hardening. Their influence on hardening comes from the rotation of grains into hard orientations. The maximum in strain hardening is caused by a significant reduction of the free path of the second pyramidal <c + a> dislocations. It is due to the formation of tension twins inside of contraction twins [17]. Molodov et al. [37,38] studied the formation of extension twins on the surface of an Mg bicrystal submitted to the simple shear test. They estimated basal slip traces in the grain interior. These slip lines continued through the twins with a small wrap at the twin boundary. A cross slip of <a> dislocations with a Burgers vector parallel to the twin boundary, was observed. The authors have shown that the slip transmission is not a local microscopic phenomenon but occurs on a macroscopic scale and may be important for the overall plastic deformation. The significance of local strains for the twin variant selection was shown in [39]. Experimental results revealed that the twin variants with higher strain accommodation factor may be activated adjacent grains across the grain boundary.

The recovery term in the Equation (11), *Θ**_r_*, depends on the temperature and strain rate [30], which indicates the participation of some thermally-activated processes. The main recovery mechanism operating in a wide temperature range is the cross slip. Atomistic models of the core structure and glide behaviour of <c + a> screw dislocations in Mg and Mg alloys found that their dissociation in both pyramidal planes of the first and second order is asymmetric giving an asymmetry in the critical resolved shear stress value. A dislocation segment in the metastable compact transition state is able to cross-slip between pyramidal planes of the second and first order [40,41,42]. Ahmad et al. analysed the double slip of <c + a> screw dislocations in pyramidal planes of the first order [43]. They found the minimum energy transition path from the pyramidal planes of the first order to the pyramidal plane of the second order and back to an adjacent pyramidal plane of the first order. Note that the analysis was performed for the Mg-Re alloys. The authors concluded that the Re solute atoms may increase the stability of the <c + a> dislocations in the pyramidal planes of the first order compared to the pyramidal planes of the second order.

### 4.2. Hall-Petch Strengthening

The TYS and CYS values plotted against *d^−1/2^*, where *d* is the grain size, are shown in Figure 10a. It is noticeable that the yield stress values were estimated higher for compression compared with the values measured for tension. This result is surprising because in textured magnesium materials the opposite tendency is usually measured, i.e., the yield stresses estimated in compression are lower than those in tension [29].

Samples prepared by rotary swaging exhibit significant internal stresses having a tensile character. This fact was confirmed in the dilatation experiments where permanent shortening of samples was observed—see Table 2. Correcting the tensile yield stress values: *TYS_C_ = TYS +*
*Δε**_p_**.E*, where *Δε**_p_* is the permanent strain and E Young‘s modulus. Analogously, for compression yield stress *CYS_C_ = CYS +*
*Δε**_p_**.E*. Taking for *E* 45 GPa, the dependences reported in Figure 10b are obtained. The slope of both dependences exhibits an identical value of *k*_ε_ = 16.4 MPa.mm^1/2^ with the correlation 0.97 for tension and 0.99 for compression. The grain size dependence of the flow stress, σ_ε_, may be expressed in the Hall-Petch relationship:(15)$$\sigma_{\varepsilon }=\sigma_{0\varepsilon \;}+\;k_{\varepsilon }\;d^{-1/2}$$
where *σ*_0ε_ is the friction stress, *k_ε_* is the microstructural stress intensity, and *d* is the average grain diameter. If dislocations are stored in pile-ups, *k*_ε_ is related to the concentration stress, *τ*_C_, as [44]:*k*_ε_ = *m*_T_ (π*m*_S_*Gb**τ*_C_/2α)^1/2^(16)
here *m*_T_ and *m*_S_ are Taylor and Sachs orientation factors, *G* the shear modulus, *b* magnitude of the Burgers vector and α a numerical factor following from the mutual dislocation interactions. Friction stresses, σ_0__ε_, measured from the dependences shown in Figure 10b, exhibit 51.8 MPa for tension and 28.6 MPa for compression. The Hall-Petch parameters estimated for tension and compression in cast Mg-2Zn alloys were analysed by Mann et al. [45]. They found a rapid increase of *k* at the beginning of plastic deformation followed by the steady state region. The values estimated for the yield stress were different for tension (14.9 MPa·mm^1/2^) and compression (17.1 MPa.mm^1/2^). With increasing strain, both parameters converged to a steady-state value of 14.2 MPa·mm^1/2^. A review of the Hall-Petch constants, *k*, found for various magnesium materials was published in [46]. Various authors studying the sensitivity to grain size estimated a spectrum of values from 2.0 up to 12.3 MPa.mm^1/2^ depending on the alloy composition, grain size interval, processing technology and the loading path. The authors of [45] concluded that the texture of samples, grain size, boundary, and temperature greatly influence *k*, in magnesium alloys. The reason consists in the deformation mode of each material and the value of the Taylor factor, which may be different depending on the texture [47]. The Taylor factor for magnesium is −4.5 for the random oriented polycrystals and decreases to 2.1–2.5 for polycrystals exhibiting a texture which inhibits the basal and prismatic slip and supports the pyramidal polyslip. In other words, the Taylor factor increases when basal or prismatic slips are dominant. As it was mentioned above, the massive twinning is observed at the beginning of plastic deformation in tension and compression. Mayers et al. [47] and Somekawa and Mukai [48] derived an analogous relationship to the Hall-Petch equation for twinning:(17)$$\sigma_{t}=\;\sigma_{0t}+k_{t}d^{-1/2}$$
where indices *t* are related to twinning. Somekawa and Mukai [49] have shown that the Hall-Petch constant *k*_t_ for twinning is higher compared to the *k*_d_ for slip deformation. The influence of the grain size on the twinning stress is much stronger [29]. Yu et al. [45] concluded that the twins’ boundaries of the extension twins represent a certain discontinuity which affects the slip of twinning transfer through the grain boundaries.

## 5. Conclusions

The AZ31 magnesium alloy was processed by rotary swaging through up to five passes. Deformation curves have been analysed with the aim to elucidate the main hardening processes occurred during tension and compression. From this work, the following conclusions may be summarised:(1)Rotary swaging developed a pronounced fibre texture.(2)The grain size decreased with stepwise swaging.(3)The strain hardening behaviour of samples is influenced by the deformation mode and texture.(4)Massive formation of twins and their interaction with dislocations caused high strain hardening in the compression tests.(5)The main softening mechanism is very probably cross-slip of <c + a> dislocations between pyramidal planes of the second and first order.(6)The yield stresses measured in compression are higher compared to the yield stress found in tension. Internal stresses originated during swaging are the reason for this behaviour.(7)Mechanical twinning increased the grain size sensitivity of the yield stress.

## Figures and Tables

**Figure 1 materials-14-00157-f001:**
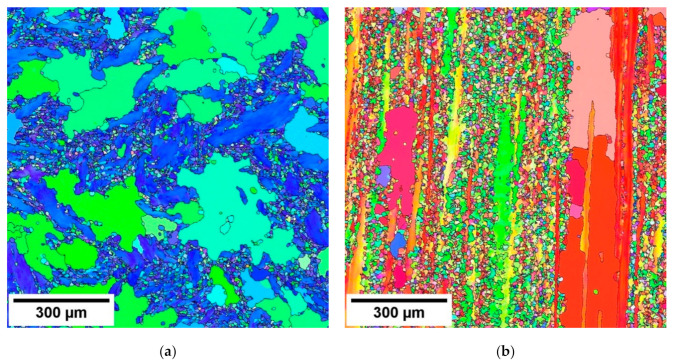
EBSD orientation maps of the extruded bar taken: (**a**) from the section perpendicular to the ED and (**b**) parallel to the ED.

**Figure 2 materials-14-00157-f002:**
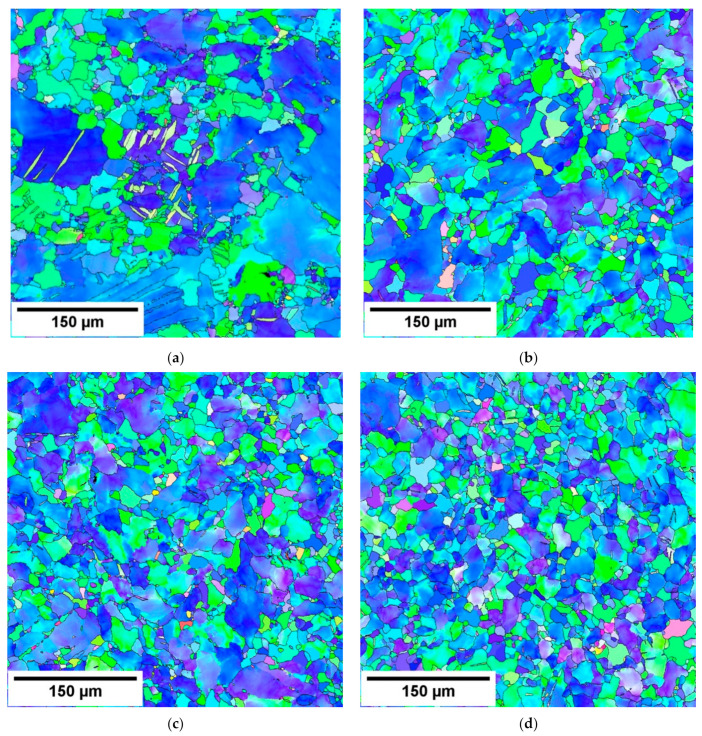
EBSD orientation maps: (**a**) SW1, (**b**) SW2, (**c**) SW3, (**d**) SW4 and (**e**) SW5.

**Figure 3 materials-14-00157-f003:**
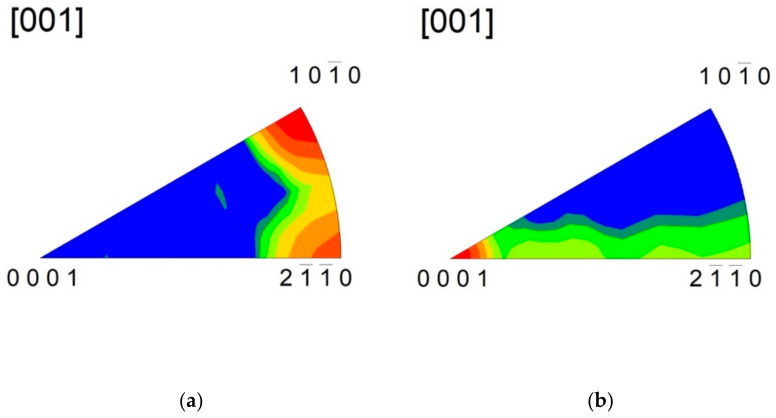
Inverse pole figures estimated for the extruded bar (SW0) taken from the section: (**a**) perpendicular to the ED and (**b**) parallel to the ED.

**Figure 4 materials-14-00157-f004:**
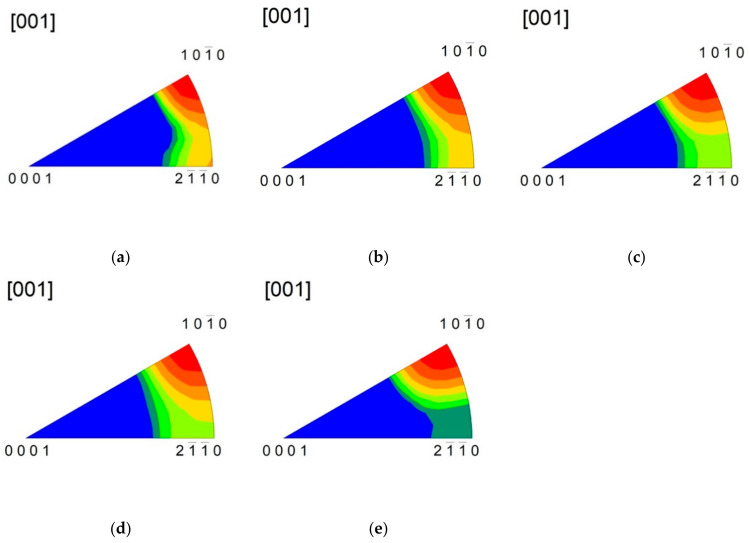
Inverse pole figures, taken from the section perpendicular to the ED, estimated for: (**a**) SW1, (**b**) SW2, (**c**) SW3, (**d**) SW4, and (**e**) SW5 samples.

**Figure 5 materials-14-00157-f005:**
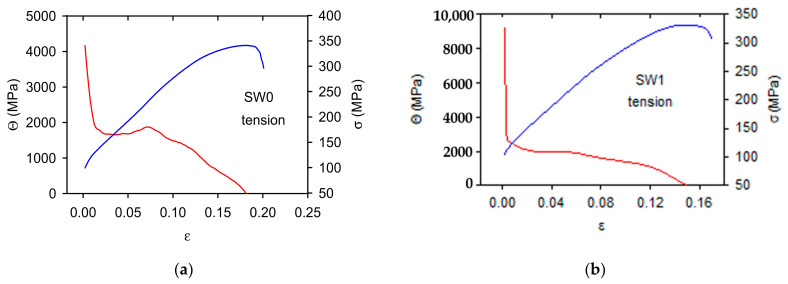
The work hardening coefficients calculated for curves obtained in tension: (**a**) SW0, (**b**) SW1, (**c**) SW2, (**d**) SW3, (**e**) SW4 and (**f**) SW5. σ-ε curves are drawn in the blue colour and Θ-ε curves in red.

**Figure 6 materials-14-00157-f006:**
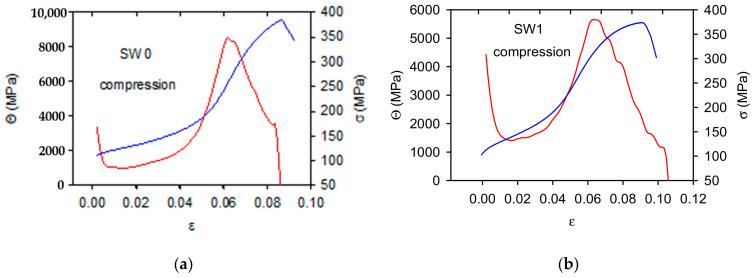
Strain dependence of the work hardening coefficients calculated for the curves obtained in compression: (**a**) SW0, (**b**) SW1, (**c**) SW2, (**d**) SW3, (**e**) SW4 and (**f**) SW5. σ-ε curves are drawn in the blue colour and Θ-ε curves in red.

**Figure 7 materials-14-00157-f007:**
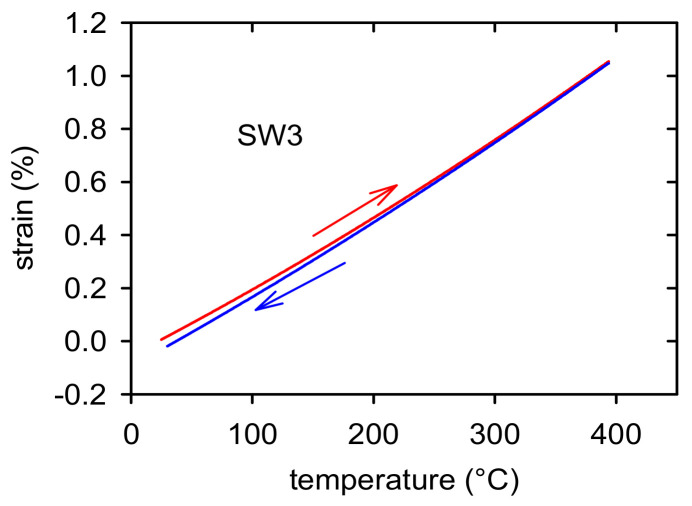
Temperature dependence of thermal expansion measured while heating and cooling for the SW3 sample.

**Figure 8 materials-14-00157-f008:**
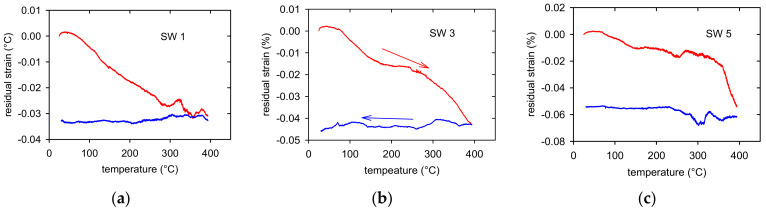
Residual strains estimated in (**a**) SW1, (**b**) SW3, and (**c**) SW5.

**Figure 9 materials-14-00157-f009:**
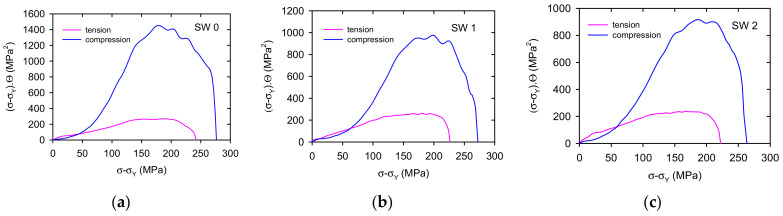
(σ − σ_Y_). Θ vs. (σ − σ_Y_) plots constructed for curves estimated in tension and compression for (**a**) SW0, (**b**) SW1, (**c**) SW2, (**d**) SW3, (**e**) SW4, (**f**) SW5.

**Figure 10 materials-14-00157-f010:**
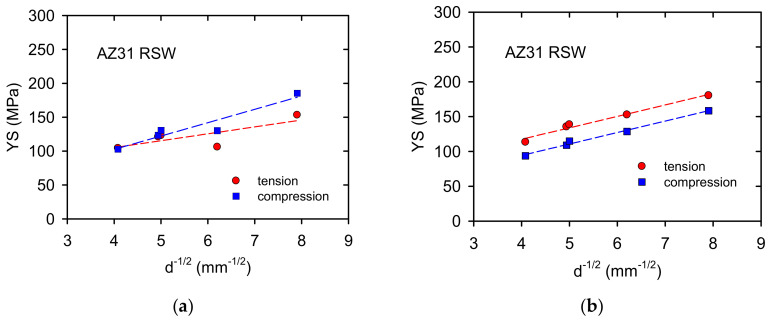
TYS and CYS depending on the grain size in the Hall-Petch plot: (**a**) measured values, (**b**) corrected values.

**Table 1 materials-14-00157-t001:** Grain size of swaged samples.

Sample	SW1	SW2	SW3	SW4	SW5
**Grain size (μm)**	60.5 ± 55.2	41.5 ± 28.6	40.3 ± 26.8	26.3 ± 13.9	16.2 ± 12.6

**Table 2 materials-14-00157-t002:** Permanent strains estimated after the first CTE measurement depending on the RSW counts.

Swaging Counts	0	1	2	3	4	5
**Permanent strain %**	0.02	0.030	0.03_5_	0.04_5_	0.05	0.06

## Data Availability

The data presented in this study are available on request from the corresponding author.

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
