# Peer review of "Strain Hardening in an AZ31 Alloy Submitted to Rotary Swaging"

_materials, 2020, doi:10.3390/ma14010157_

Round 1

Reviewer 1 Report

The paper deals with investigation of deformation of a magnesium alloy. Extensive experiments are a valuable contribution of this work. The paper contains some interesting experimental data. On the other hand, there are several weak points in the manuscript which have to be explained:

  1. Introduction is not convincing. The whole introduction is focused on discussion of various publications, which contributed to development of material models based on evolution of dislocation populations, beginning from fundamental works of Kocks, Estrin and Mecking to more advanced solutions. However, all these papers are more than 25 years old. The recent papers are not discussed. The introduction neither gives motivation for the present work nor shows unexplored areas which would allow to formulated the objectives.
  2. There are several cumulative citations in the Introduction. Hundreds of papers on modelling evolution of dislocations were published. Only the most important papers should be cited and each of them should be separately discussed with emphasis on its new contribution. The cumulative citations are useless. Beyond this, the justification, why one parameter model was used in the present work, is too weak.
  3. The new contribution of the paper is not clear. The whole introduction is focused on dislocation density model. Only the last sentence in the introduction states briefly, that comparison of strain hardening in tension and compression is the objective of the paper. The whole introduction does not refer to papers dealing with physical simulations of magnesium alloys.
  4. All investigated phenomena are thermally activated. Therefore, constraining the experiments to one temperature and one strain rate is not justified. The influence of the temperature and strain rate on twinning is important.
  5. The first paragraph in the chapter 4 contains discussion of earlier papers, mainly Authors’ papers. Following this, without any connecting sentence equations describing evolution of dislocation populations are presented. All these equations are well known from the literature. It seems that statement that the work hardening rate is much higher for compression samples is the main output of the chapter “Discussion”. This observation is well known.
  6. There is no information on how the twinning can be accounted for in equations describing evolution of dislocation populations. This aspect is mentioned in the paper but the precise approach is not given. The only influence is in equation (12), where contribution of twinning to the average free path for dislocations is introduced. Is it the only contribution of twinning?
  7. Difficulty to distinguish between discussion of earlier publications, including Authors’ publications, and a new contribution of the present paper, is the main weak point of the work which does not allow me to accept it for publication.
  8. The conclusions are not constructive. These are statements, which are commonly known
  9. Some symbols in equations are not explained.
  10. English is acceptable, although it might be improved. Syntax should be changed in some sentences.

Author Response

Dear reviewer, please see attached file. Thank you very much for your work.

Reviewer 2 Report

Good work.

Graphic images often don't have the same scale (I.E Figure 5, 6, 8 and 9),
this makes it difficult to compare the results.

A few words should be spent in the introduction to describe the Rotary Swaging Process.

Author Response

(The authors gave the same response as above.)

Reviewer 3 Report

I think it's a well-written paper. The conclusions drawn are useful from an academic and practical point of view. Please add and correct with the following points.

1) Unify the terms used throughout the paper. For example, rotary swaging is unified with RSW. In addition, there are several redundant expressions in the present manuscript. Be scrutinized again.

2) Chapter 2: How many times did the tension and compression experiments be performed under the same conditions?

3) Chapter 2: How did you find Θ (derivative) from the σ-ε curve in Fig. 5? Did you approximate the σ-ε curve, or did you use the secant method? Please be specific.

4) Chapter 2: Describe the length measurement method and accuracy of the sample used in the residual stress and residual strain measurement.

5) Page 5, lines 146-147: Insert the sentence "as shown in Table 1".

6) Page 6, lines 153-155: It says {???̅?}, but isn't it {2110}?

7) Indicate clearly the σ-ε and Θ-ε curves within Figures 5 and 6.

8) There is no scale on the vertical axis in Figure 5 (c).

9) Page 12, line 273: Correct equation (12) to equation (14).

Author Response

(The authors gave the same response as above.)

Reviewer 4 Report

Materials-1009393

The present manuscript describes the strain hardening behavior in magnesium AZ31 alloys submitted to rotary swaging.

The manuscript in well-organized and written, clear and coincise. I have only some remarks in order to improve the overall quality of the paper.

Therefore, I suggest to publish it onto Materials after minor revisions.

1) some data about the supplier of AZ31 extruded rods should be added.

2) Table 2: the accuracy of the data reported (0.032-0.035 vs. 0.06) should be revised.

3) overlapping images of the analyses performed in the case of Figure 5 and 5 should be added.

Author Response

(The authors gave the same response as above.)

Round 2

Reviewer 1 Report

The paper was noticeably improved. All comments in my review were accounted for. The paper can be accepted in the present form.